# Impact of a Hybrid Assisted Wheelchair Propulsion System on Motion Kinematics during Climbing up a Slope

**Bartosz Wieczorek \***, **Łukasz Warguła** and **Dominik Rybarczyk**

Department of Mechanical Engineering, Poznan University of Technology, 60-965 Poznań, Poland;
lukasz.wargula@put.poznan.pl (Ł.W.); dominik.rybarczyk@put.poznan.pl (D.R.)

**\*** Correspondence: bartosz.wieczorek@put.poznan.pl; Tel.: +48-(61)-665-20-42

**Abstract:** Overcoming terrain obstacles presents a major problem for people with disabilities or with limited mobility who are dependent on wheelchairs. An engineering solution designed to facilitate the use of wheelchairs are assisted-propulsion systems. The objective of the research described in this article is to analyze the impact of the hybrid manual–electric wheelchair propulsion system on the kinematics of the anthropotechnical system when climbing hills. The tests were carried out on a wheelchair ramp with an incline of 4°, using a prototype wheelchair with a hybrid manual–electric propulsion system in accordance with the patent application P.427855. The test subjects were three people whose task was to propel the wheelchair in two assistance modes supporting manual propulsion. The first mode is hill-climbing assistance, while the second one is assistance with propulsion torque in the propulsive phase. During the tests, several kinematic parameters of the wheelchair were monitored. An in-depth analysis was performed for the amplitude of speed during a hill climb and the number of propulsive cycles performed on a hill. The tests performed showed that when propelling the wheelchair only using the hand rims, the subject needed an average of 13 ± 1 pushes on the uphill slope, and their speed amplitude was 1.8 km/h with an average speed of 1.73 km/h. The climbing assistance mode reduced the speed amplitude to 0.76 km/h. The torque-assisted mode in the propulsive phase reduced the number of cycles required to climb the hill from 13 to 6, while in the climbing assistance mode the number of cycles required to climb the hill was reduced from 12 to 10 cycles. The tests were carried out at various values of assistance and assistance amplification coefficient, and the most optimally selected parameters of this coefficient are presented in the results. The tests proved that electric propulsion assistance has a beneficial and significant impact on the kinematics of manual wheelchair propulsion when compared to a classic manual propulsion system when overcoming hills. In addition, assistance and assistance amplification coefficient were proved to be correlated with operating conditions and the user's individual characteristics.

**Keywords:** wheelchair; hybrid manual–electric drives; drives supporting the movement

## 1. Introduction

The main problem faced by disabled people using wheelchairs are terrain obstacles, either architectural or natural ones, such as thresholds, hills, or surface irregularities. All these factors translate into increased resistance during wheelchair propulsion. To facilitate wheelchair propulsion for people with disabilities, scientists and inventors are developing innovative design solutions for suspension systems, propulsion systems, etc. An innovation in the field of wheelchair propulsion systems are wheelchairs with electric–manual hybrid propulsion systems that allow people with

disabilities to manually propel the wheelchair and use the support of the drive torque generated by electric motors. From the experience of users of such wheelchairs, most of them consider this solution to increase their mobility and social integration [1]. It allows them to increase their traveling distance [2], and facilitates tasks requiring increased torque on the propulsion wheels [3].

Despite their advantages, hybrid propulsion systems also have drawbacks, such as: difficulty to transport in a vehicle [1,3], limited battery life [1], limited use inside the home [3], the need to take into account control delay characterized by unnatural interaction with the user [3–5]. What is particularly noticeable is reduced precision of control compared to classic wheelchairs [3], due to the increased weight, size and the possibility of delay of control signals [6]. In addition, there is a risk that users of wheelchairs with assisted-propulsion systems might lead a less active lifestyle limiting their physical development, which may, in turn, predispose them to many long-term health problems [6]. An important aspect when designing and using wheelchairs with propulsion assistance systems is the use of control algorithms. Literature on the subject presents practical and effective algorithms for interference suppression to ensure safe and comfortable operation [7]. These algorithms can detect and suppress external forces not related to the propulsion by human hands [7]. Oh and Hori (2013) describe algorithms that respond to gravitational forces acting on slopes [7]. Research is also being carried out on optimal cross-coupling control strategies, which were applied to a rim motor wheelchair, with the aim of adjusting the speed ratio of the left and right wheels in order to follow specified paths with various turning radius [8]. In general, one can distinguish the concept of assisted-propulsion control, which assumes increasing the torque applied to the wheels by the user [9].

Van der Woude et al. [6] with his team point out that propulsion assistance technology supports mobility and plays an important role in the design of wheelchairs in terms of: wheelchair mechanics, the mobility of the user, and the interface of the wheelchair user. The design used in the research according to the patent application P.427855 [10] described in the book "Research on the biomechanics of manual wheelchair drive for innovative manual and hybrid drive" [11] equipped with additional propulsion systems, is consistent with modern wheelchair design trends. The modular structure of the developed propulsion system makes it possible to adapt it to most mass-produced wheelchairs. Available propulsion modes such as: purely manual or electric propulsion systems and hybrid modes, i.e., hill-climbing assistance and assistance with torque in the propulsive phase make the described design innovative and consistent with contemporary design trends. The functions of the wheelchair hybrid propulsion system are consistent with the description of trends in wheelchair development technology indicated by Cooper and his team [12]. These trends relate to the increased safety of use, efficiency, accessibility and, above all, increased mobility of disabled people [12]. Research on people with limited mobility has shown that people whose mobility was improved by the use of robotic prostheses, exoskeletons, and electric wheelchairs [13] claimed greater satisfaction with the quality of their life than people with lower mobility [14–16].

The efficiency and adaptation to the needs of the user of the algorithms developed to assist wheelchair propulsion is closely dependent on the operating conditions of the wheelchair. This finding highlights the importance of experimental research describing the conditions, as well as the parameters of the used designs. To analyze the conditions, researchers use wheel speed, the angle of inclination, and external force values, which Oh et al. [17] and his team indicate as the most important physical quantities for controlling the power-assisted wheelchair (PAW) in everyday conditions of use [17].

The authors have developed their own hybrid manual–electric propulsion assistance systems because testing innovative control and design algorithms requires fully recognized and program-edited components. The use of available constructions to test new concepts is difficult due to unadaptable controllers or an individual control program known only to designers. SmartDrive MX + [18] is one of the commercial drivers for wheelchairs with a propulsion drive. The main disadvantage of the algorithms developed with its use is not adapting to changing conditions, only setting constant supportive torque settings. In the solution developed by the authors of the hybrid wheelchair, the kinematics of the movement of the upper limbs is analyzed and on this basis the values of the

supporting moment are set. In addition to the control unit itself, the hybrid wheelchair described by the authors differs from the other ones described in the literature, the sensors used in the control algorithm. The solution described by Oh and Hori [7] did not use inertial sensors in the form of an accelerometer and gyroscope allowing direct recognition of the slope, and indirect methods were used to recognize it [7]. In another construction described by Oh et al. (2014) the control algorithm required the use of torque meters in pushrims [17]. Similar to a design developed by Cooper et al. (2002), which was characterized using additional mechanical gears [9]. The hybrid cart developed by the authors of the article similar functionality was obtained without the use of torque meters, and it was replaced by an analysis of kinematics of upper limb movement measured by incremental encoders.

This article examines the results of research on the impact of hybrid manual–electric propulsion assistance systems of wheelchairs on the kinematics of the anthropotechnical system while overcoming an obstacle—a wheelchair ramp with a slope of 4°. To do that, the impact of the system and its settings on operating functions, vehicle speed and the number of pushes was estimated as compared to classic wheelchair propulsion with the use of hand rims. Two modes were tested: hill-climbing assistance and amplification of the driving torque generated by the upper limb. In the first mode, the system controller used an accelerometer and a gyroscope. The second mode was used to assist movement when propelling hand rims. In this mode, when a change in the rotational speed of the drive wheel was detected, the controller set the current value proportional to the value of the assistance amplification coefficient. The main objective of the research was to verify the hybrid propulsion control system and experimental selection of the assistance amplification coefficient for selected wheelchair operating conditions in order to find an optimal value to maintain a balance between reducing physical effort and maintaining the precision of wheelchair control with the use of hand rims.

## 2. Materials and Methods

The research was conducted on three adults. Table 1 presents information on the subjects. The subjects were physically fit, which was dictated by safety considerations. A new prototype design was tested, for which the limit values of electric motor torque amplification of the driving torque generated by the human upper limb were determined. Full fitness of the tested subjects increased the likelihood of avoiding an accident in the event of loss of control over the tested wheelchair. The selection criteria of the examined people assumed that patients must be physically fit, all people selected for the examination had a similar level of physical condition and experience in using the wheelchair. The selection criteria of the examined persons assumed that it must be physically fit, all persons selected for examination have a similar level of physical condition, and the level of experience in using the wheelchair must be at a similar level. During the classification of the examined subjects, their height, weight, age, circumferential force exerted on the pushrim during its repulsion and experience with using a wheelchair were taken into account. The experience with using a wheelchair was reflected by the number of points (from 1 to 5).

**Table 1.** Comparison of anthropometric features and the level of experience in wheelchair operation of the test subjects, where push force is a circumferential force applied to the pushrims generated by the upper limb.

| Anthropometric Features and the Level of Experience | Height cm | Weight kg | Age years | Push Force N | Experience |
|:---:|:---:|:---:|:---:|:---:|:---:|
| Subject 1 | 176 | 72 | 33 | 180 | ●●●○○ |
| Subject 2 | 170 | 96 | 32 | 170 | ●●●●○ |
| Subject 3 | 180 | 108 | 29 | 200 | ●●○○○ |

Implementation of the research carried out by a team led by B. Wieczorek, was approved by the Bioethics Committee of the Karol Marcinkowski Medical University in Poznan directed by P. Chęciński, by Resolution No. 1100/16 of November 10, 2016. The test subjects were familiarized with the test procedures and gave their written consent to participate in it and to publish the results of the research. The classification carried out for the purposes of the research does not raise any ethical concerns.

Measurement tests were carried out on a straight track consisting of three sections. On the first horizontal section, the wheelchair was accelerated, on the middle one, inclined at an angle, the kinematics were analyzed, and on the last horizontal section, the wheelchair was stopped (Figure 1). The track was located indoors without any atmospheric effects on the kinematics of the wheelchair [19]. The track surface was made of oak floorboards. Based on previous tests, the rolling resistance coefficient for this surface was determined at 0.028–0.029 [20].

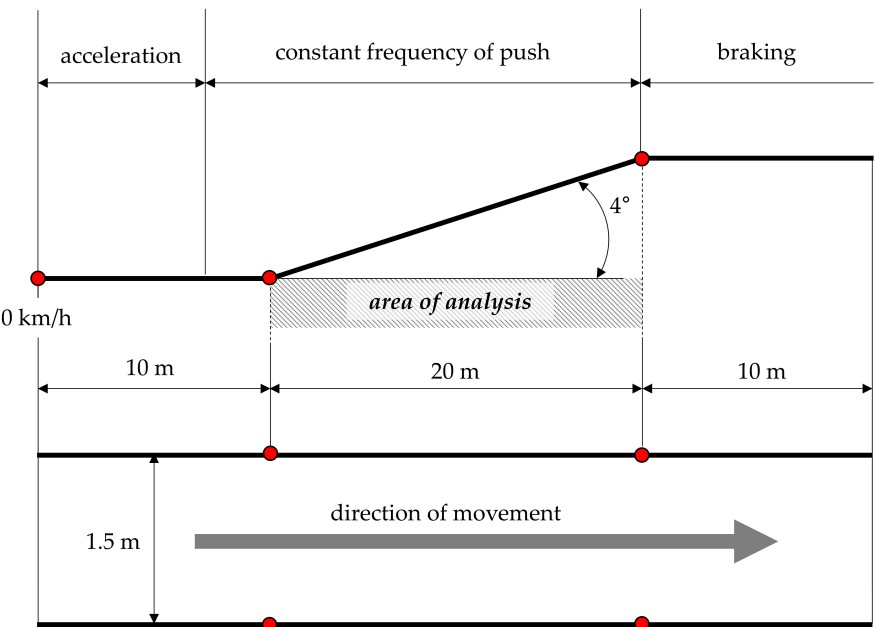

**Figure 1.** Dimensions of the track used in the research.

The tests were carried out for an ARMedical AR-405 wheelchair with a cross frame equipped with a prototype hybrid propulsion system module (Figure 2). The wheelchair modified in this way had two Golden Motor BLCD MagicPie 5 motors (BrushLess Direct-Current motors), with a power of 500W built in the hubs of the drive wheels (a), incremental encoders (b), an original control system with gyroscope (c) and a touch screen controller (d). The screen was used to select the assistance mode and the value of the assistance amplification coefficient w, which can be set smoothly from 0 to 100%. The wheelchair control system was made based on a 32-bit STM32F407 microcontroller operating with a frequency of 100 MHz (Figure 3). The main purpose of the control system was to control the rotational speed of two BLDC motors connected to the wheel rims. The speed was registered by means of two incremental encoders connected to configured digital inputs, in the counter mode. The signal was counted in a quadrature mode, which increased the measurement accuracy while also allowing determination of the direction of rotation. The speed value was determined with a basis of 30 ms using an independent counter.

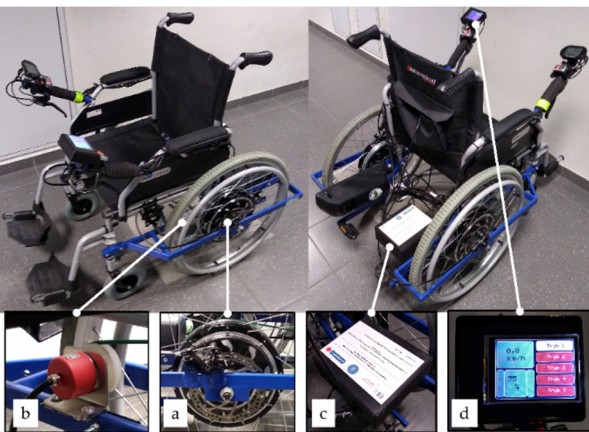

**Figure 2.** Tested AR-405 wheelchair equipped with a hybrid propulsion module. (**a**) BrushLess Direct-Current motor; (**b**) incremental encoder; (**c**) electronic controller; (**d**) touch screen.

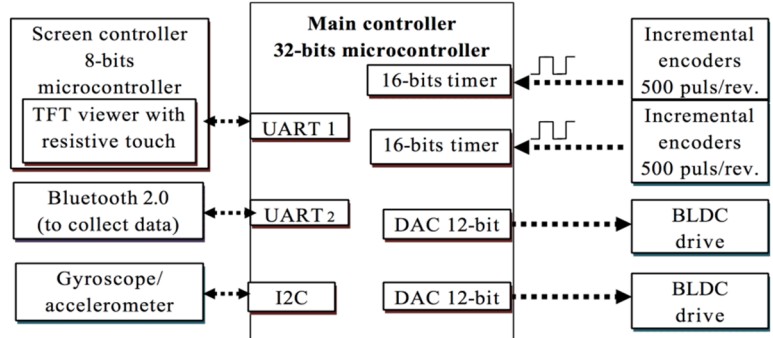

**Figure 3.** Block diagram of the control system.

In the research, two modes were tested: hill-climbing assistance (t3) and amplification of the driving torque (t4) generated by the upper limb. In the first mode, the system controller used an accelerometer and a gyroscope. The acceleration values measured by the accelerometer and angular velocity were converted to angular position with the use of a complementary filter. In addition, the acceleration values were corrected using a low-pass filter. The measured values were then sent to the controller, where the amplification coefficient $w$ was added. Due to vibrations generated during movement, especially when moving on uneven surfaces (e.g., cobblestone) and a sharp tilt of the wheelchair due to inertia forces, the system was equipped with two independent software angle measurement tools. The first checked if the value of the angular position was greater than 1° for a time period longer than 300 ms. Only then was movement assistance initiated. The second one verified if the angular position was below 1° for a time period longer than 300 ms. If it was, assistance was switched off. The value of the voltage controlling the motor for the hill-climbing assistance mode was determined using the following Formula (1):

$$u = \alpha \cdot c + r_3 \cdot w + \delta \tag{1}$$

where: $u$—signal given to the BLDC motor controller, $\alpha$—wheel chair angle [°], $c$—angle coefficient (0.0075) $r_3$—gain coefficient in mode 3 (0.0016), $w$—assistance amplification coefficient [%], $\delta$—voltage offset (1.102) [V].

The amplification of the driving torque mode (t4) was used to assist propulsion with the use of hand rims. When a change in the rotational speed of the drive wheel was detected, the controller set the current value proportional to the value of the assistance amplification coefficient $w$. The control

was carried out with a time base of 100 ms. The voltage controlling the motor for this mode was determined using the following Formula (2):

$$u = r_4 \cdot w + \delta \tag{2}$$

where: $u$—signal given to the BLDC motor controller, $r_4$—gain coefficient in mode 4 (0.008), $w$—assistance gain coefficient [%], $\delta$—voltage offset (1.102) [V]. Four values of the assistance amplification coefficient were tested. The list of controlling voltages for these values is shown in Table 2.

**Table 2.** Motor controlling voltage for the amplification coefficient values tested.

| w [%] | t3 ($\alpha = 4°$) u [V] | t4 u [V] |
|---|---|---|
| 25 | 1.1875 | 1.3025 |
| 50 | 1.2275 | 1.5025 |
| 75 | 1.2675 | 1.7025 |
| 100 | 1.3075 | 1.9025 |

The test procedure consisted of each subject moving four times along the designated track with five assistance amplification coefficient settings (0%, 25%, 50%, 75%, 100%) for each of the two assisted modes tested. The user had a 10-minute break to rest between each test. The subjects were asked to accelerate the wheelchair on the first horizontal section, and then to maintain a constant rhythm of pushing the hand rims on the slope. Then, on the last horizontal section of the track, they had to brake the wheelchair to a standstill.

The data measured with the use of incremental encoders was used to analyze the kinematics of the wheelchair. The analyzed track included only the section where the wheelchair was moving uphill. During the climbing up the slope, the patients repelled the pushrims with an interval equal to 1 s. The interval value was chosen experimentally. For the examined patients, the assumed length of the interval prevents the wheelchair from rolling down. Based on the data collected for each measurement test, graphs of the speed and acceleration of the wheelchair were prepared (Figure 4). Analyzing the measured waveforms of speed and acceleration, the end points of the propulsive cycle (PC) were detected. Based on these points, the total number of propulsive cycles (CC), the number of propulsive cycles during the climb uphill (CH), the average speed of the wheelchair on the hill $M\,v$, the maximum speed amplitude during the climb $\Delta v$, and the average acceleration of the wheelchair $M\,a$ were determined.

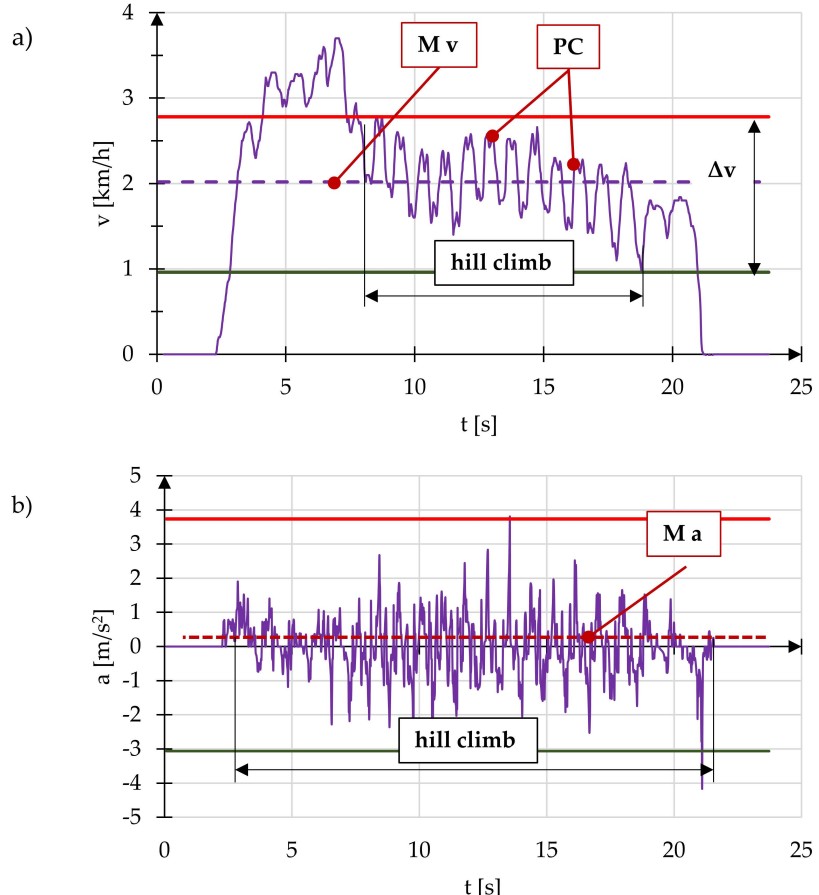

**Figure 4.** Examples of the graphs analyzed in the tests: speed (**a**) and acceleration (**b**); PC: the end points of the propulsive cycle, *M v*: the average speed of the wheelchair on the hill, *M a*: the average acceleration of the wheelchair.

In addition, after the measurements were carried out, the test subjects were interviewed. They had to evaluate their control over the wheelchair and the sensation during power-assisted movement depending on the value of the amplification coefficient on a 5-point scale. The purpose of the survey was to compare the subjective perceptions of the subjects with the kinematic parameters obtained during the measurements.

## 3. Results

During the tests, the impact of the type of assistance mode and the assistance amplification coefficient $w$ selected for it on the speed, acceleration of the wheelchair, and the number of propulsive cycles required to overcome the hill were analyzed. According to the adopted research methodology, 5 settings of the assistance amplification coefficient 0%, 25%, 50%, 75% and 100% were tested. For the $w = 0\%$ setting, the wheelchair had no assistance, while for the $w = 100\%$ setting, the motor assisted the manual propulsion system with the maximum motor torque. For one of the subjects, the coefficient setting $w = 100\%$ was not tested, as the subject refused to take part in this particular test, as they were afraid of losing control of the wheelchair. Speed waveforms for a wheelchair without electric power assistance are shown in Figure 5. The selected speed waveform for hill-climbing assistance mode (t3) is shown in Figure 6. In turn, Figure 7 shows the speed waveform for amplification of the driving torque mode (t4). On the waveforms, the starting points for the hill climb (SC), the starting points for the assistance mode (SA) and the end points for the hill climb (END) are marked.

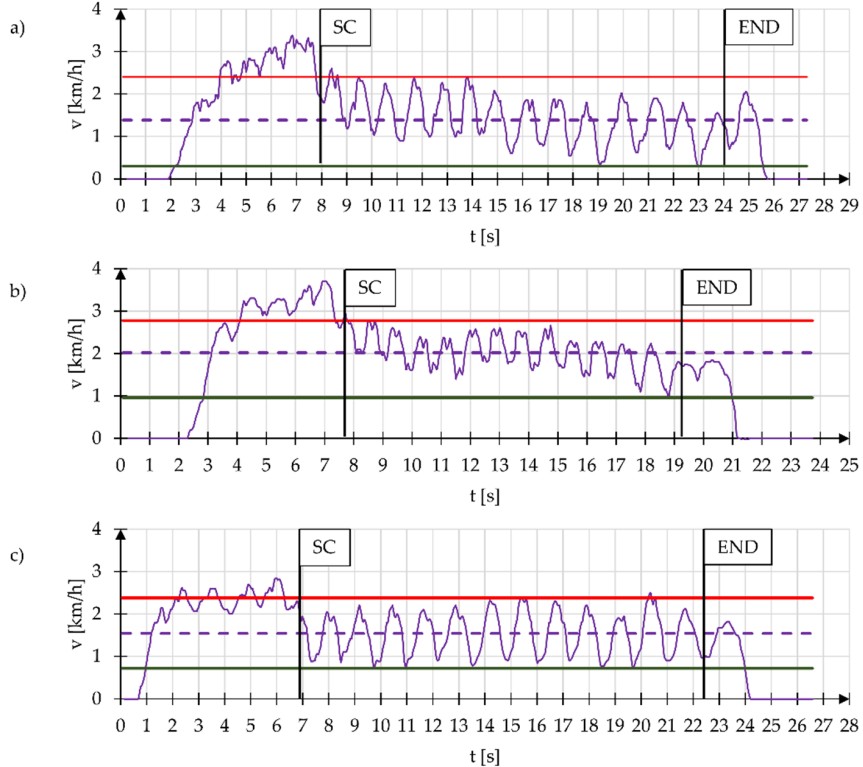

**Figure 5.** Speed waveforms for a wheelchair moving without electric motor assistance for subject 1 (**a**), subject 2 (**b**) and subject 3 (**c**), where SC – the starting points for the slope climb, END – the end points for the slope climb.

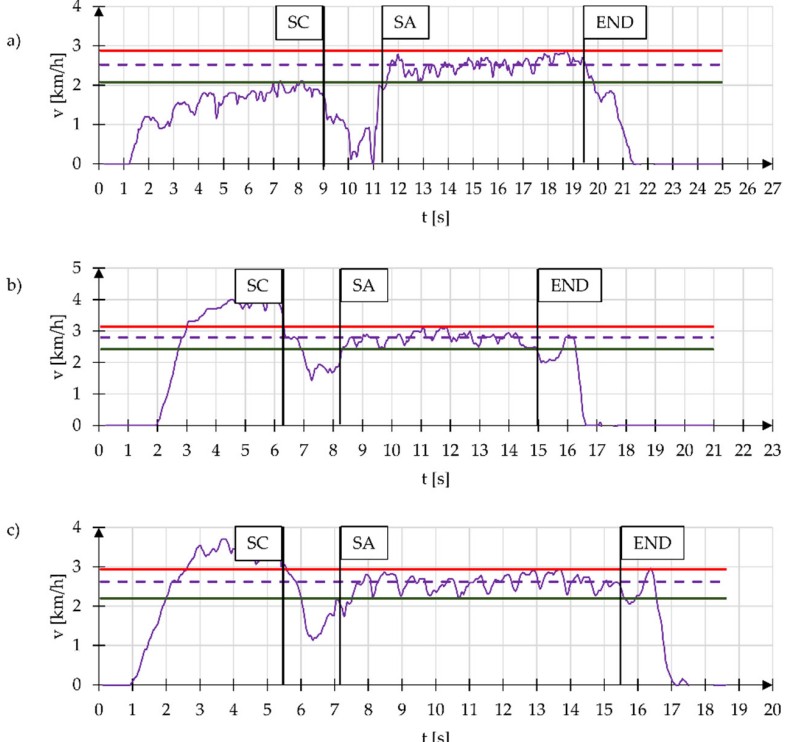

**Figure 6.** Speed waveforms of a wheelchair for Hill Climb Assistance Mode (t3) with amplification coefficient $w = 50\%$ for subject 1 (**a**), subject 2 (**b**) and subject 3 (**c**), where SC – the starting points for the slope climb, SA – the starting points for the assistance mode, END – the end points for the slope climb.

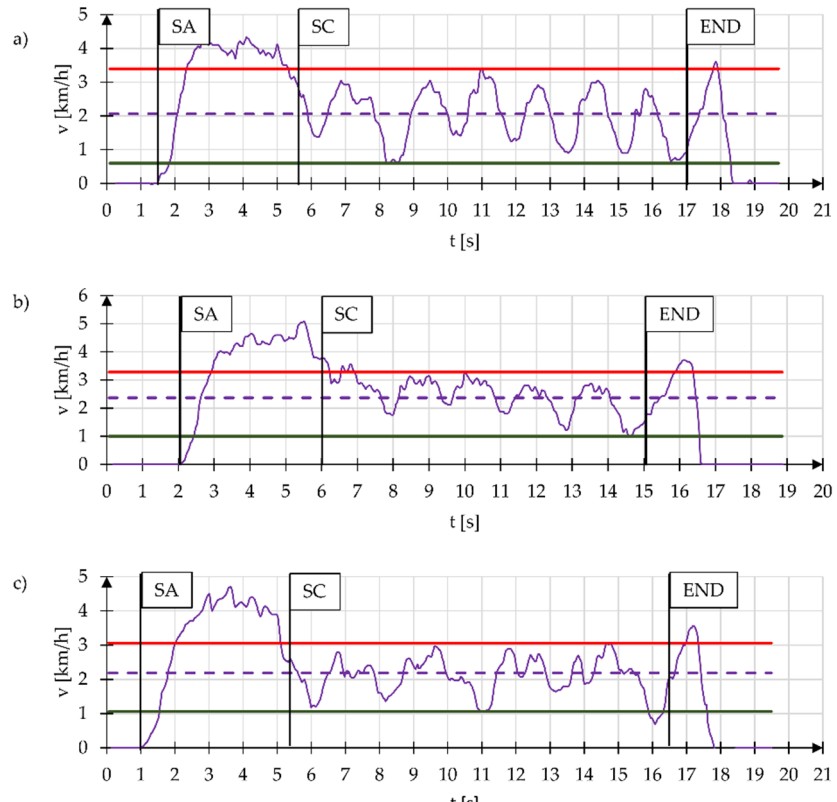

**Figure 7.** Speed waveforms of a wheelchair for the Torque Amplification Mode (t4) with the amplification coefficient $w$ = 50% for subject 1 (**a**), subject 2 (**b**) and subject 3 (**c**).

To corroborate the similarity of the measured speed waveforms for individual subjects, Pearson's correlation analysis was performed. For mode t3, the value of the average correlation coefficient for measuring the speed amplitude $\Delta v$ was 98%, and the average correlation coefficient for measuring the number of propulsive cycles on the CH hill was 95%. For mode t4, the value of the average correlation coefficient for measuring the speed amplitude $\Delta v$ was 99%, and the average correlation coefficient for measuring the number of propulsive cycles on the CH hill was 83%. Tables 3 and 4 show the average results of the analyzed parameters for the examined subjects, which include the total number of CC propulsive cycles, the number of propulsive cycles during the uphill motion, average speed of the uphill climb $M v$, speed amplitude on the hill $\Delta v$ and the average acceleration during moving uphill $M a$. Figures 8 and 9 show the amplitude of wheelchair speed when moving uphill as a function of the assistance amplification coefficient for modes: hill-climbing assistance (t3) and amplification of the driving torque (t4).

**Table 3.** List of average values of the analyzed kinematic parameters for the hill-climbing assistance mode (t3).

| w | Kinematic Parameters | Subject 1 | Subject 2 | Subject 3 | Average | SD |
|---|---|---|---|---|---|---|
| | CC | 19 ± 1 | 22 ± 1 | 19 ± 0 | 19 | 1 |
| | CH | 12 ± 1 | 15 ± 0 | 12 ± 0 | 12 | 1 |
| 0% | M v [km/h] | 2.03 | 1.31 | 1.51 | 1.85 | 0.26 |
| | Δv [km/h] | 1.70 | 1.95 | 1.87 | 1.76 | 0.14 |
| | M a [m/s²] | −0.004 | −0.005 | −0.003 | −0.004 | 0.006 |
| | CC | 16 ± 1 | 22 ± 2 | 18 ± 1 | 17 | 1 |
| | CH | 11 ± 0 | 14 ± 0 | 11 ± 0 | 11 | 0 |
| 25% | M v [km/h] | 2.34 | 2.25 | 2.32 | 2.33 | 0.10 |
| | Δv [km/h] | 0.81 | 0.88 | 0.86 | 0.83 | 0.08 |
| | M a [m/s²] | −0.010 | 0.009 | 0.008 | −0.004 | 0.012 |

**Table 3.** *Cont.*

| w | Kinematic Parameters | Subject 1 | Subject 2 | Subject 3 | Average | SD |
|---|---|---|---|---|---|---|
| | CC | 14 ± 1 | 20 ± 1 | 17 ± 2 | 15 | 2 |
| | CH | 10 ± 1 | 12 ± 1 | 11 ± 0 | 10 | 1 |
| 50% | M v *[km/h]* | 2.77 | 2.56 | 2.64 | 2.73 | 0.08 |
| | Δv *[km/h]* | 0.70 | 0.71 | 0.88 | 0.76 | 0.16 |
| | M a *[m/s²]* | −0.008 | −0.004 | −0.003 | −0.006 | 0.010 |
| | CC | 13 ± 0 | 18 ± 0 | 16 ± 0 | 14 | 1 |
| | CH | 9 ± 1 | 9 ± 1 | 10 ± 1 | 9 | 1 |
| 75% | M v *[km/h]* | 2.88 | 2.78 | 2.94 | 2.90 | 0.06 |
| | Δv *[km/h]* | 0.71 | 0.98 | 0.85 | 0.75 | 0.16 |
| | M a *[m/s²]* | −0.001 | 0.008 | −0.015 | −0.006 | 0.010 |
| | CC | 11 ± 1 | 19 ± 1 | 13 ± 1 | 12 | 1 |
| | CH | 7 ± 2 | 9 ± 0 | 9 ± 1 | 7 | 2 |
| 100% | M v *[km/h]* | 3.19 | 3.43 | 3.47 | 3.28 | 0.14 |
| | Δv *[km/h]* | 0.75 | 0.70 | 0.69 | 0.73 | 0.13 |
| | M a *[m/s²]* | −0.001 | 0.008 | −0.002 | −0.001 | 0.026 |

**Table 4.** Average values of analyzed kinematic parameters for amplification of the driving torque mode (t4).

| w | Kinematic Parameters | Subject 1 | Subject 2 | Subject 3 | Average | SD |
|---|---|---|---|---|---|---|
| | CC | 19 ± 1 | 22 ± 1 | 19 ± 1 | 20 | 2 |
| | CH | 12 ± 1 | 15 ± 1 | 12 ± 0 | 13 | 1 |
| 0% | M v *[km/h]* | 2.01 | 1.31 | 1.51 | 1.61 | 0.31 |
| | Δv *[km/h]* | 1.70 | 1.95 | 1.87 | 1.84 | 0.18 |
| | M a *[m/s²]* | −0.006 | −0.005 | −0.003 | −0.005 | 0.008 |
| | CC | 19 ± 2 | 17 ± 0 | 18 ± 0 | 18 | 1 |
| | CH | 12 ± 1 | 12 ± 0 | 12 ± 1 | 12 | 1 |
| 25% | M v *[km/h]* | 1.66 | 1.41 | 1.63 | 1.57 | 0.19 |
| | Δv *[km/h]* | 1.61 | 1.95 | 1.84 | 1.80 | 0.21 |
| | M a *[m/s²]* | −0.007 | 0.006 | −0.005 | −0.002 | 0.024 |
| | CC | 10 ± 1 | 10 ± 1 | 8 ± 1 | 9 | 1 |
| | CH | 5 ± 0 | 6 ± 0 | 6 ± 1 | 6 | 1 |
| 50% | M v *[km/h]* | 2.28 | 1.98 | 2.14 | 2.13 | 0.15 |
| | Δv *[km/h]* | 2.08 | 2.91 | 2.29 | 2.43 | 0.45 |
| | M a *[m/s²]* | −0.043 | −0.017 | 0.011 | −0.017 | 0.025 |
| | CC | 5 ± 1 | 5 ± 0 | 6 ± 1 | 5 | 1 |
| | CH | 2 ± 0 | 3 ± 0 | 4 ± 1 | 3 | 1 |
| 75% | M v *[km/h]* | 3.02 | 2.07 | 2.62 | 2.57 | 0.49 |
| | Δv *[km/h]* | 2.81 | 4.28 | 3.15 | 3.41 | 0.79 |
| | M a *[m/s²]* | 0.014 | −0.120 | 0.017 | −0.030 | 0.071 |
| | CC | 4 ± 0 | n/a | 4 ± 1 | 4 | 1 |
| | CH | 2 ± 0 | n/a | 2 ± 0 | 2 | 0 |
| 100% | M v *[km/h]* | 3.13 | n/a | 3.37 | 3.25 | 0.16 |
| | Δv *[km/h]* | 3.83 | n/a | 4.23 | 4.03 | 0.30 |
| | M a *[m/s²]* | 0.023 | n/a | 0.121 | 0.072 | 0.093 |

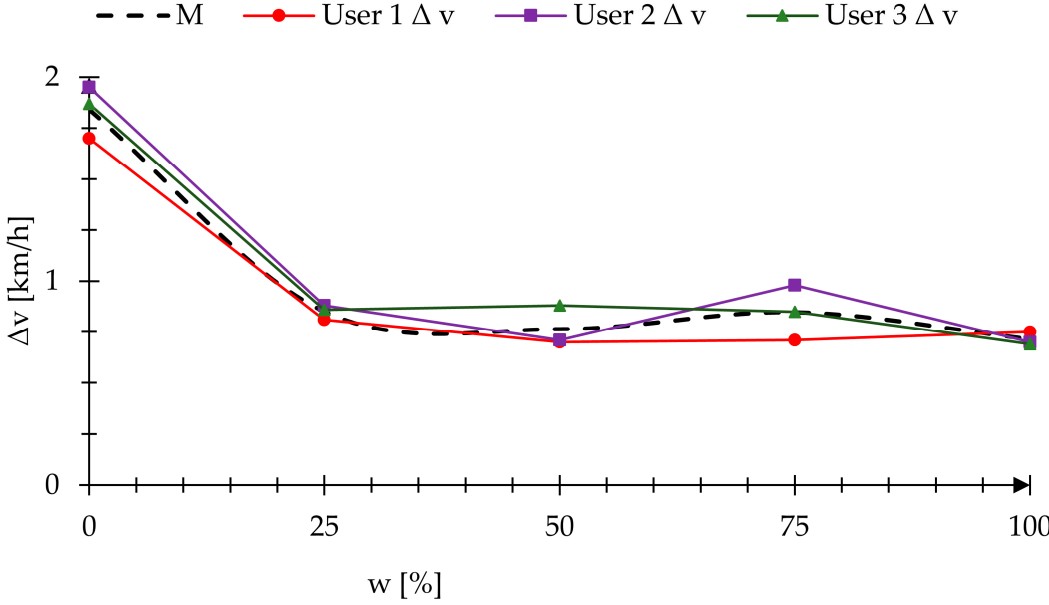

**Figure 8.** Graphs showing the maximum speed amplitude as a function of the assistance amplification coefficient during the climb uphill as a function of the assistance amplification coefficient for the hill-climbing assistance mode (t3).

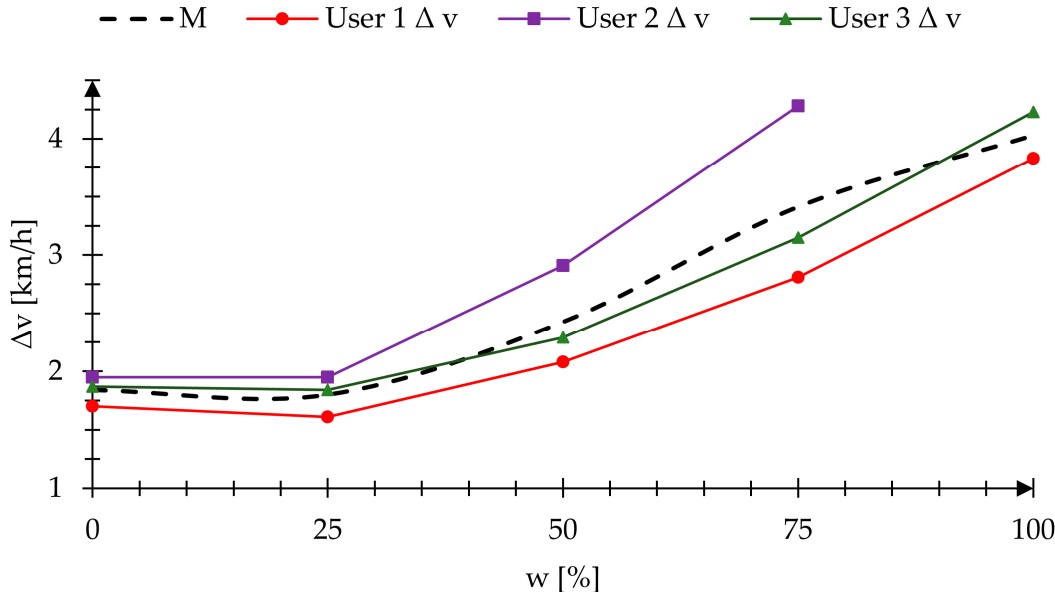

**Figure 9.** Graphs showing the maximum speed amplitude as a function of the assistance amplification coefficient during the climb uphill as a function of the assistance amplification coefficient for amplification of the driving torque mode (t4).

According to the adopted research methodology, each subject had to fill out a questionnaire after finishing one mode of the test trial. Table 5 presents the assessment of individual assistance modes.

**Table 5.** The results of the questionnaire conducted among the test subjects on the effort during wheelchair propulsion and the degree of control over the wheelchair depending on the value of the assistance amplification coefficient.

| w | value | t3 | | | t4 | | |
|---|---|---|---|---|---|---|---|
| | | Subject 1 | Subject 2 | Subject 3 | Subject 1 | Subject 2 | Subject 3 |
| 0% | effort | ●●●●● | ●●●●● | ●●●●● | ●●●●● | ●●●●● | ●●●●● |
| | control | ●●●●● | ●●●●● | ●●●●● | ●●●●● | ●●●●● | ●●●●● |
| 25% | effort | ●●●●○ | ●●●●○ | ●●●○○ | ●●●●● | ●●●●● | ●●●●○ |
| | control | ●●●●● | ●●●●● | ●●●●● | ●●●●● | ●●●●○ | ●●●●○ |
| 50% | effort | ●●●○○ | ●●●○○ | ●●○○○ | ●●●●○ | ●●●●○ | ●●●○○ |
| | control | ●●●●○ | ●●●●○ | ●●●○○ | ●●●○○ | ●●●○○ | ●●○○○ |
| 75% | effort | ●●○○○ | ●●○○○ | ●○○○○ | ●●●○○ | ●●●○○ | ●●●○○ |
| | control | ●●●○○ | ●●●○○ | ●●●○○ | ●●○○○ | ●●○○○ | ●○○○○ |
| 100% | effort | ●○○○○ | ●○○○○ | ○○○○○ | ●●●○○ | n/a | ●●○○○ |
| | control | ●○○○○ | ●○○○○ | ○○○○○ | ●○○○○ | n/a | ○○○○○ |
| | Key: ●●●●● – maximum, ○○○○○ – minimum | | | | | | |

## 4. Discussion

The tests were carried out on three subjects with different levels of wheelchair propulsion skills and different techniques of propulsion using hand rims [21]. Therefore, it was first verified whether the measurements obtained for these subjects can be compared, using the Pearson correlation coefficient. The obtained values for the measured parameters showed a strong correlation between the number of drive cycles performed for individual patients. The correlation coefficient was more than 90% between individual patients, regardless of the value of the assistance amplification coefficient. Based on that, an assumption was made that difference in skills of the subjects does not exclude the possibility of comparing among them the basic parameters of motion kinematics of the wheelchair.

Analyzing the speed waveforms during the uphill climb without the use of assistance modes ($w = 0\%$), an average speed value $M v$ equal to 1.73 km/h on the hill was noted. The obtained average speed result is consistent with the results of other researchers. The publication [22] showed that an unmodified track on an incline of approximately 3° reaches an average speed of 1.54 km/h. Additionally, when analyzing the measurements, a characteristic increase in the speed amplitude on the slope was observed compared to the movement of the wheelchair on the horizontal section. The average speed amplitude $\Delta v$ on the hill was 1.8 ± 0.14 km/h. Due to low average uphill speed and large speed amplitude, the subjects needed an average of 12 ± 1 full propulsive cycles to reach the hill. The results obtained for the system without assistance are obvious, and result from a significant increase in the motion resistance, being the effect of the force of gravity, which is confirmed by the impact on the kinematics of wheelchair movement also by other researchers. The assistance modes were used to reduce the impact of rolling resistance forces on wheelchair motion kinematics. The optimal configuration of the assistance mode control program requires the selection of appropriate values of the assistance amplification coefficient. The selected value should noticeably improve kinematic parameters, reduce the user's effort and allow control over the wheelchair, guaranteeing safety of use.

Analyzing the hill-climbing assistance mode (t3), it was observed that regardless of the value of the assistance amplification coefficient (w), the BLCD motors supporting the drive torque generated by the upper limb started start with a delay, which was shown on the speed waveforms between SC and SA. The delay was caused by the motor control program, which, in the first 300 ms after changing the angle, checks if the wheelchair is on the ramp or the angle change read by the gyroscope was temporary due to uneven terrain. For each tested subject, the delay in the start of the assistance relative to the time the wheelchair was on the slope was on average 2 sec. It is possible to reduce the time during which the control program detects the wheelchair inclination angle. However, that would make

the wheelchair more sensitive to accidental interference, and would cause automatic activation of the hill-climbing assistance mode, especially on uneven terrain that generates high vibration.

Analyzing the speed of a wheelchair moving with the hill-climbing assistance mode turned on (t3), it was determined that between the points SC and SA, i.e., before starting the BLCD motors, the speed of the wheelchair was at a similar level as during the climb uphill without the assistance modes. However, after passing the SA points, a significant improvement in the kinematic parameters of the wheelchair was noted. For the value $w = 25\%$, the speed amplitude $\Delta v$ was reduced by more than half to 0.83 km/h. For the remaining values of the assistance amplification coefficient ($w = 25$–75%), the speed amplitude remained constant within the range of 0.73–0.83 km/h. In addition, the performed tests proved that as the assistance amplification coefficient increased, the number of propulsive cycles required to climb the hill decreased. For $w = 0\%$, 12 propulsive cycles were required and for $w = 100\%$ - 7, although, according to the test subjects, excessively high value of the amplification coefficient resulted in a more difficult control of the wheelchair when maneuvering it using hand rims. Based on the test subject questionnaires, as well as an analysis of the change in speed amplitude as a function of the assistance amplification coefficient, an optimal value of the coefficient was selected at 50%, which meant that the voltage controlling the motor had a value of 1.2275 V. For such settings, a balance was maintained between the reduction of muscle effort and manual control over the wheelchair.

The operation of the amplification of the driving torque mode (t4) did not cause any delays in the control system caused by the slope detection, since this mode was not initiated by a detection of the wheelchair inclination. The t4 mode was started when the upper limb began to propel the drive wheel, which was measured with incremental encoders. As a result, this assistance mode was active already on the initial horizontal section of the track. Analyzing the kinematics of the wheelchair, it was determined that increasing the assistance amplification coefficient increases the average wheelchair speed on the ramp $M v$. This average for $w = 0\%$ was $1.61 \pm 0.31$ km/h, while for $w = 100\%$ it doubled and equaled $3.25 \pm 0.16$ km/h. A similar tendency was observed for the speed amplitude on the hill $\Delta v$. The speed amplitude for the coefficient $w = 25\%$ remained constant at 1.80-1.84 km/h. However, for the remaining values of the assistance amplification coefficient, it increased, reaching $\Delta v = 4.03 \pm 0.30$ km/h for $w = 100\%$. The problem of increasing amplitude resulted from the adopted algorithm that required wheelchair users to initiate wheel movement to activate support by the BLCD motors. Therefore, in this mode, assistance is turned off at the end of each propulsive cycle and is started as soon as the next one starts. The increase of the $w$ coefficient translated to a larger torque generated by the electric motor.

In mode t4, the impact of an increase in the assistant amplification coefficient on a decrease in the number of propulsive cycles required for a hill climb was also observed. For $w = 0\%$ the number of CH cycles was $13 \pm 1$, and for $w = 100\%$ it was only 2. In this mode as well, the test subjects found that higher values of the assistance amplification coefficient resulted in difficult control over manual maneuvering of the wheelchair. In addition, as was demonstrated in the subject questionnaires, even the highest assistance amplification coefficient does not significantly reduce the effort required to propel the wheelchair during a hill climb. Based on the collected data, it was found that in order to maintain a balance between the reduction of muscle effort and manual control of the wheelchair, the value of the assistance amplification coefficient should not be greater than 50%, which means the voltage controlling the electric motor should be lower than 1.5025 V.

In addition, for each of the analyzed assistance modes, the average acceleration of the wheelchair on the slope was measured. Analysis how this average value changed provided qualitative data on the tendency of the wheelchair movement. When the average $\Delta a$ approached zero, it meant that the wheelchair was traveling at a constant speed, and when the average moved away from zero, becoming negative, it could be assumed that the wheelchair was slowing down along the traveled path. Analysis of this variable allowed determination of when the torque generated by electric motors was interfering with the propulsive movements generated by the upper limbs.

## 5. Summary

The conducted tests allowed the researchers to determine that the use of hybrid propulsion systems has a positive impact on the kinematics of the wheelchair and the human body. Regardless of the assistance mode used, the number of propulsive cycles decreased as the assistance amplification coefficient increased. Fewer propulsive cycles, in turn, meant reduced muscle effort, as well as maximum values of muscle activity when propelling a wheelchair [23–27]. However, for the analyzed track, mode t3 seemed to have more advantages, as it was initiated as soon as the wheelchair inclination was detected and continued operating until the wheelchair was again on a horizontal surface. As a result, resistance resulting from gravitational forces on a hill was reduced, letting the wheelchair user feel as if moving on a horizontal surface in terms of effort. In mode t3, excessively high value of the assistance amplification coefficient caused its automatic motion up the hill, eliminating participation of the upper limbs in the process. This translated into difficult control over the wheelchair with the use of hand rims.

A characteristic feature of the amplification of the driving torque mode (t4) is that it can be used both on horizontal surfaces and on hills. Assistance in this mode is started when the user initiates propulsive motion. In the case of moving uphill, this meant that in the initial stages of pushing the hand rims, the subject had to overcome all resistance forces with the use of their muscle strength, which resulted in short, but intense, strain on the muscular system. The need to propel the hand rims to initiate the assistance mode meant large speed amplitudes, especially for larger values of the assistance amplification coefficient. The observed reduction in the number of propulsive cycles during the uphill climb was a result of a rapid acceleration of the wheelchair after the initiation of the propulsive cycle. This rapid acceleration significantly interfered with wheelchair control and steering.

Based on the conducted test, it was found that for the examined subjects the best results in terms of improvement of the wheelchair kinematics were obtained for the assistance amplification coefficient ranging from 25 to 50%. With such value settings, the subjects could clearly feel the impact of the electric propulsion system on the manual one, while at the same time, maintaining full control over the wheelchair trajectory. The conclusions drawn from the tests allowed verification of the control system algorithm and select the appropriate values for the voltage controlling the electric motors. It should be noted that the selected values of the assistance amplification coefficient only apply to hard surfaces with a low rolling resistance coefficient. The tests were performed only on three patients; however, this amount allowed determination of the value of the assist gain coefficient needed. It should be noted that the work was carried out on a prototype and focuses on developing a preliminary version of the control software.

Further tests will be carried out for other types of surfaces. This will make it possible for a user of the described wheelchair prototype to select one of three buttons representing different types of urban surfaces instead of setting the percentage of the assistance amplification coefficient. Such an approach will increase the number of factors that integrate the human with their wheelchair, and allow effective control over propulsion assistance [28,29].

**Author Contributions:** Conceptualization, B.W., Ł.W. and D.R.; methodology, B.W., Ł.W. and D.R.; software, D.R.; validation, B.W., Ł.W. and D.R.; formal analysis, B.W., Ł.W. and D.R.; investigation, B.W., Ł.W. and D.R.; resources, B.W., Ł.W. and D.R.; data curation, B.W., Ł.W. and D.R.; writing—original draft preparation, B.W., Ł.W. and D.R.; writing—review and editing, Ł.W; visualization, B.W.; supervision, B.W.; project administration, B.W.; funding acquisition, B.W. All authors have read and agreed to the published version of the manuscript.

**Funding:** The test was performed as part of the project LIDER VII "Testing the manual wheelchair propulsion biomechanics for innovative manual and hybrid propulsions" (LIDER/7/0025/L-7/15/2016) financed by the Polish National Centre for Research and Development.

**Conflicts of Interest:** The authors declare no conflict of interest.

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
