# Peer review of "Impact of a Hybrid Assisted Wheelchair Propulsion System on Motion Kinematics during Climbing up a Slope"

_applsci, doi:10.3390/app10031025_

Round 1

Reviewer 1 Report

The paper in its current format is better than the first version but still needs some major modifications

Replacing “Hill climb” for “terrain obstacles” still does not adequately represent the experimental protocol. The study specifically evaluates climbing up slope/ramp. For reference, in the USA a very standard ramp (recommended by American’s with disabilities Act) has a slope of around 4.8 degrees. In the introduction section Sundaram et al (2017) and Ding & Cooper (2005) papers talk about power wheelchairs. These are motorized wheelchairs which are operated using a joystick or equivalent electronic control mechanism. The type of system that the authors have developed and tested in this study improve the propulsion efficiency of wheelchairs that are manually propelled. These are fundamentally different wheelchairs than power wheelchairs. It does not make sense to use references for power wheelchairs to justify the need for hybrid propulsion systems. The whole first paragraph needs to be re written with appropriate references. References at other places need to be verified as well. The introduction section is anyways very long. The readability can be significantly improved by making this section concise and specific to the technology developed for this study and by removing extra references. One essential element lacking in the introduction is the rationale for developing this specific hybrid propulsion system. What are drawbacks in the currently available products that serve the same purpose? How different are these control algorithms compared to those in the existing products? What problems in the existing hybrid propulsion systems do the authors anticipate to address using the algorithms that are developed for this study? The Abstract reports reduction in propulsion cycles only in one mode and not in the other. It is recommended to be consistent in the reporting the outcome measures when comparing modes. Line 125: Similar to comment from last round of review – the specific criterion used to “classify” participants based on height, weight, age, experience with WC should be clearly mentioned. If researchers evaluated more than 3 participants then whey were the other candidates excluded and only these 3 selected? How were these participants recruited? Random sampling or convenience sampling? Figure 1: Authors report the total propulsion cycles. So the ramp section along cannot be the area of analysis. It is expected to be the whole ramp. Since propulsion cycles and speed of wheelchair are important outcome measures it is important to specify what instructions the participants were given before starting the trial. Were the participants instructed to “go as fast as possible” or to “go at a comfortable pace”. How can the authors be sure that every participant exerted an equivalent amount of power (force*displacement/time)? Although there may not be an easy solution to measuring and controlling the individual’s force applied during propulsion, authors could assume that the participants have similar abilities and giving a standard instruction could lead to them applying equivalent amount of force. This technique is common in wheelchair propulsion studies. In addition, standardized tools/measures could be used to evaluate upper extremity strength in the participants and reported along with the demographics information. Line 241 mention that you are talking about CH Table 2 needs a correct caption Table 4 needs appropriate column headings Line 271- a new result is introduced in the beginning of the discussion section. It is unclear which outcome measures were correlated between the participants. Were correlations run separately for each value of w? Try to report std deviations wherever average values are reported (e.g. line 294) A common practice is to use the discussion section to compare results/metrics from your study with similar published studies by other researchers, if authors could find any. For example, authors could find average speeds wheelchair users achieve while climbing an ADA standard ramp.   There may be some major limitations in study design that could be identified from this review. For example, small sample size is a limitation for this study and making a selection of a major control factor in the control algorithm from 3 participants may not be recommended. Authors should use the discussion section to acknowledge possible study design limitations like this and describe their implications i.e. how they might have affected (or not affected) their major results. This is a common practice implemented while reporting human subject studies.

Author Response

Author's Reply to the Review Report (Reviewer 1)

Reviewer: 1

Dear Reviewer,

we would like to thank you very much for your comments. We have modified the paper according to your remarks, as follows.

Authors

Replacing “Hill climb” for “terrain obstacles” still does not adequately represent the experimental protocol. The study specifically evaluates climbing up slope/ramp. For reference, in the USA a very standard ramp (recommended by American’s with disabilities Act) has a slope of around 4.8 degrees.

Answer: We agree with the remark, The title of the article has been corrected.

In the introduction section Sundaram et al (2017) and Ding & Cooper (2005) papers talk about power wheelchairs. These are motorized wheelchairs which are operated using a joystick or equivalent electronic control mechanism. The type of system that the authors have developed and tested in this study improve the propulsion efficiency of wheelchairs that are manually propelled. These are fundamentally different wheelchairs than power wheelchairs. It does not make sense to use references for power wheelchairs to justify the need for hybrid propulsion systems. The whole first paragraph needs to be re written with appropriate references. References at other places need to be verified as well. The introduction section is anyways very long. The readability can be significantly improved by making this section concise and specific to the technology developed for this study and by removing extra references.

Answer: We have modified the paper according to your remarks.

One essential element lacking in the introduction is the rationale for developing this specific hybrid propulsion system. What are drawbacks in the currently available products that serve the same purpose? How different are these control algorithms compared to those in the existing products? What problems in the existing hybrid propulsion systems do the authors anticipate to address using the algorithms that are developed for this study? The Abstract reports reduction in propulsion cycles only in one mode and not in the other. It is recommended to be consistent in the reporting the outcome measures when comparing modes.

Answer: We have modified the paper according to your remarks.

The Abstract reports reduction in propulsion cycles only in one mode and not in the other. It is recommended to be consistent in the reporting the outcome measures when comparing modes.

Answer: We have modified the paper according to your remarks.

Line 125: Similar to comment from last round of review – the specific criterion used to “classify” participants based on height, weight, age, experience with WC should be clearly mentioned. If researchers evaluated more than 3 participants then whey were the other candidates excluded and only these 3 selected? How were these participants recruited? Random sampling or convenience sampling?

Answer: In accordance with the recommendations from the last round of reviews, we have added a table (Table 1.) with detailed information about patients. The selection of only three patients resulted from a limited group of available people. The criteria for selection assumed that the person must be physically fit, all persons selected for the examination had a similar level of physical condition, and the level of experience in using the wheelchair had to be at a similar level. Physical fitness was required for safety reasons. Users' physical condition was assessed on the basis of other tests using a wheelchair ergometer and surface electromyography. The level of experience was due to the time spent in a wheelchair on the other subjects.

Information of the selection criteria for the examine patients was added to the manuscript.

Figure 1: Authors report the total propulsion cycles. So the ramp section along cannot be the area of analysis. It is expected to be the whole ramp.

Answer: The article analyzes the total number of drive cycles (CC) and the number of drive cycles performed only on the ramp (CH). The total number of cycles includes the first level sub-recess, ramp and number of cycles performed on the second horizontal section until the wheelchair stops. The number of drive cycles performed on the ramp contained only the inclined section. At the final analysis, only a 20-meter section of the ramp was taken into account. Because in the case of the initial horizontal and final sections, discrepancies between the driving styles of the subjects were noted. In the case of the section on which the subjects were accelerating, despite adopting the same frequency of performing drive cycles, different angles of rotation of the wheel were observed while repelling the strings. However, on the last horizontal stretch, patients braked immediately after driving on it or after performing several drive cycles. In contrast to these two horizontal sections, only on the ramp (probably due to an increase in resistance to movement), patients performed similar angles of wheel rotation when pushing back thrusts, in addition their upper limb always followed the same path referred to in literature as the "arc". (Boninger, M. L., Souza, A. L., Cooper, R. A., Fitzgerald, S. G., Koontz, A. M., & Fay, B. T. (2002). Propulsion patterns and pushrim biomechanics in manual wheelchair propulsion. Archives of physical medicine and rehabilitation, 83(5), 718-723.)

Since propulsion cycles and speed of wheelchair are important outcome measures it is important to specify what instructions the participants were given before starting the trial. Were the participants instructed to “go as fast as possible” or to “go at a comfortable pace”.

Answer: In the case of the test, the required wheelchair pace was due to the resistance to movement resulting from the uphill climb. It had to be such that the cart did not roll down. Based on preliminary studies not included in the article, it was experimentally determined that three selected patients with their upper limb strength should perform motive motions at a frequency of 1 second.

Interval information has been added to the manuscript text

How can the authors be sure that every participant exerted an equivalent amount of power (force*displacement/time)? Although there may not be an easy solution to measuring and controlling the individual’s force applied during propulsion, authors could assume that the participants have similar abilities and giving a standard instruction could lead to them applying equivalent amount of force.

Answer: We are unable to determine whether the study participants exerted the same power in the classic definition of power. However, on the occasion of previous experiments, we have developed a power equivalent factor for anthropechnical systems coupling the human body with a wheelchair. This factor is the quotient of muscle activity and distance traveled by the wheelchair in one drive cycle. Muscle activity was examined by surface electromyography.

Considering this factor and the same procedures for climpbing the wheelchair on a slope, we assumed that the people selected for the study had similar physical abilities.

This technique is common in wheelchair propulsion studies. In addition, standardized tools/measures could be used to evaluate upper extremity strength in the participants and reported along with the demographics information.

Answer: As recommended, we performed additional tests in which we defined the average repelling force of the strings. The tests were performed using a dynamometer additionally attached to the pushrims. The measured data was added to the table with demographic data.

Line 241 mention that you are talking about CH Table 2 needs a correct caption

Answer: We have modified the paper according to your remarks.

Table 4 needs appropriate column headings

Answer: Table 4 has the same headers as Table 3 because the same parameters are reported.

Line 271- a new result is introduced in the beginning of the discussion section. It is unclear which outcome measures were correlated between the participants. Were correlations run separately for each value of w?

Answer: Added explanation in the text.

Try to report std deviations wherever average values are reported (e.g. line 294)

Answer: We have modified the paper according to your remarks.

A common practice is to use the discussion section to compare results/metrics from your study with similar published studies by other researchers, if authors could find any. For example, authors could find average speeds wheelchair users achieve while climbing an ADA standard ramp.

Answer: We have modified the paper according to your remarks

There may be some major limitations in study design that could be identified from this review. For example, small sample size is a limitation for this study and making a selection of a major control factor in the control algorithm from 3 participants may not be recommended. Authors should use the discussion section to acknowledge possible study design limitations like this and describe their implications i.e. how they might have affected (or not affected) their major results. This is a common practice implemented while reporting human subject studies.

Answer: We have modified the paper according to your remarks.

Reviewer 2 Report

The paper analyses the impact of a hybrid assisted wheelchair propulsion system on motion kinematics during hill climbs. The topic fits the scope of the journal well. The paper might have value for science and general practice, however I have some remarks which should be taken into account and incorporated in the paper before publication.

Main issues:

1. Please define “value of assistance” and “assistance amplification coefficient” at their first occurrence in the text.

2. Please clearly define all operational modes and unify their markings and names. In the present form it is difficult to follow the text, e.g. in the text you use: mode t3, t4, mode 3, first mode/hill climbing assistance, second mode / assistance with propulsion torque in the propulsive phase, torque assisted mode in the propulsive phase, “propulsion modes such as: purely manual or electric propulsion systems and hybrid modes, i.e. hill climbing assistance and assistance with torque in the propulsive phase make the described design innovative and consistent with contemporary design trends.” etc. I suggest to present it in the form of a table.

3. Please expand all abbreviations at their first occurrence in the text.

Technical and formal issues:

4. A list of all symbols and abbreviations used in the paper would improve the quality of communication.

5. I suggest to remove scientific titles, e.g. “Ph.D., Eng.,”

6. Table 1 – please add a unit for “age”.

7. Figure 1 – “area” -> “subject”.

8. Figure 5 – please describe all the abbreviations.

9. Lines 179, 187 – “offset” -> “voltage offset”.

10. Table 5 – please do not finish a section with a table. Provide an additional discussion.

11. Line 189 – “Tables should be placed in the main text near to the first time they are cited.” To be removed.

12. Table 3 – please provide a clear description of its context, e.g. is “M v” a “v max”? Please use appropriate markings.

13. Line 250 – What does a “test trial” mean?

14. Please improve the quality of figures and provide appropriate legends were applicable.

Author Response

Author's Reply to the Review Report (Reviewer 2)

Reviewer: 2

Dear Reviewer,

we would like to thank you very much for your comments. We have modified the paper according to your remarks, as follows.

Authors

The paper analyses the impact of a hybrid assisted wheelchair propulsion system on motion kinematics during hill climbs. The topic fits the scope of the journal well. The paper might have value for science and general practice, however I have some remarks which should be taken into account and incorporated in the paper before publication.

Main issues:

Please define “value of assistance” and “assistance amplification coefficient” at their first occurrence in the text.

Answer: We have modified the paper according to your remarks.

Please clearly define all operational modes and unify their markings and names. In the present form it is difficult to follow the text, e.g. in the text you use: mode t3, t4, mode 3, first mode/hill climbing assistance, second mode / assistance with propulsion torque in the propulsive phase, torque assisted mode in the propulsive phase, “propulsion modes such as: purely manual or electric propulsion systems and hybrid modes, i.e. hill climbing assistance and assistance with torque in the propulsive phase make the described design innovative and consistent with contemporary design trends.” etc. I suggest to present it in the form of a table.

Answer: We have modified the paper according to your remarks.

Please expand all abbreviations at their first occurrence in the text.

Answer: We have modified the paper according to your remarks.

Technical and formal issues:

A list of all symbols and abbreviations used in the paper would improve the quality of communication.

Answer: We agree with your opinion, however, we are not sure if it is compatible with the format of the journal, so we made sure that all symbols were explained in the text. In addition, we avoided descriptions containing the symbols themselves without explaining them in the sentence.

I suggest to remove scientific titles, e.g. “Ph.D., Eng.,”

Answer: We have modified the paper according to your remarks.

Table 1 – please add a unit for “age”.

Answer: We have modified the paper according to your remarks.

Figure 1 – “area” -> “subject”.

Answer: We used the word "area" because we refer in this drawing to a certain surface of the track on which we performed the analysis, and not to the patient as the examined object.

Figure 5 – please describe all the abbreviations.

Answer: We have modified the paper according to your remarks.

Lines 179, 187 – “offset” -> “voltage offset”.

Answer: We have modified the paper according to your remarks.

Table 5 – please do not finish a section with a table. Provide an additional discussion.

Answer: This is because of the size of images and embedding them in the text, so as to avoid blank pages.

Line 189 – “Tables should be placed in the main text near to the first time they are cited.” To be removed.

Answer: We have modified the paper according to your remarks.

Table 3 – please provide a clear description of its context, e.g. is “M v” a “v max”? Please use appropriate markings.

We explained in the paragraphs preceding the tables that the "M v" designation is the average uphill speed.

Line 250 – What does a “test trial” mean?

Answer: The "test trial” is one wheelchair ride along the described track (in the adopted test methodology).

Please improve the quality of figures and provide appropriate legends were applicable.

Answer: We have increased Image resolution to 300 DPI.

Round 2

Reviewer 1 Report

Table 1: Please add brief description about "Push Force". 

Column headers of Table 5 are still unchanged and say "Subject 1". Did the authors intend to say Subjects 1, 2 3? 

Line 100: Word "similar" appears twice in the sentence

Author Response

Dear Reviewer,

we would like to thank you very much for your comments. We have modified the paper according to your remarks, as follows:

Table 1: Please add brief description about "Push Force".

Answer: We've added a description.

Column headers of Table 5 are still unchanged and say "Subject 1". Did the authors intend to say Subjects 1, 2 3?

Answer: We have modified the paper according to your remarks.

Line 100: Word "similar" appears twice in the sentence

Answer: We have modified the paper according to your remarks.

This manuscript is a resubmission of an earlier submission. The following is a list of the peer review reports and author responses from that submission.

Round 1

Reviewer 2 Report

The paper presents a preliminary evaluation of a hybrid wheelchair propulsion system using a small sample. The paper presents design and performance of certain propulsion assistance modes. The paper reads more like a design report one would put together after developing certain a new technology/algorithms. Although the paper reports data on human participants, it does not appear as a good research report because of the several methodological gaps in the study design and discussion.

The abstract mentions that these tests were conducted on terrain obstacles. The study only included 2 level surfaces and ramps which had a “smooth and even” surface. Calling this minimal setup as terrain obstacle seems like a stretch. One rationale for developing this system, as mentioned by the authors, is that providing assistance with propulsion helps develop muscle strength and improves blood circulation. This is a big claim and has to be justified with adequate citations. Demographics (age, weight, height, gender), skills level, mobility limitations and disabilities (if any), recruitment methods (including random sampling or convenience sampling) for the study sample are not mentioned. These help readers to get context and helps explain why certain results are the way they are. It is only in the discussion section that the authors mention that the 3 participants had different levels of propulsion skills. It is ideally recommended to test technologies with the actual end users of certain technology. Having data from existing manual wheelchair users will make this paper strong. Complete set of dimensions of the track used in the study are missing. The overall research hypotheses and/or objectives of the human subject trails are missing. What’s the hypothesis authors were trying to address/what were their expectations when modifying the w - assistance gain coefficient? Did they consider asking the participants which w they preferred? There is no systematic participant interview data presented in the paper. There are few discussion points that mention interviewing participants but methods section do not describe the techniques and questionnaires used and results section don’t present the interview results. It is good to see representative graphs of the potential propulsion sessions figures 2-4 but authors don’t identify the participants that are performing the selected trials. Are those from the same person or different people? Table 1 title needs to be changed to represent what is shows Authors mention that the motor control system adapted to kinematic forces applied to the pushrims. It would be nice to see how the forces differed for the different conditions that were tested. It was not clear how the total number of trials was 36. 3 participants x 3assistance modes x 4 repetitions is 36 trials but then authors show data from trials that varied assistance gain (w). How and when were these trials conducted? It is recommended to have a random sequence of gain and assistance modes to avoid carry over effects between two subsequent trials. Tables 2 -4 don’t seem to be necessary. It is harder for the reader to identify trends in the data from a long numerical table. Graphs will be significantly better in displaying the same information. Ideally, the same information could have been provided using means and std deviations for different categories of data. The tables 5-6 also seem unnecessary. It is unusual to see the correlations that are displayed in the tables. They may be okay for checking the validity of data but have not seen such correlations reported in scientific papers. In legends of Figure 5 and 6 use a full word instead of “M” The discussion and summary sections barely describe the results of the study including explanations for any of the major limitations of the study. It is also customary to compare your results to results that are already published and this gives validity to your own work. The first sentence of the conclusion talks about navigating terrain obstacles. The simple track that was used in this study does not qualify for a terrain obstacle. Please consider using person first language throughout the paper as a sign of respect to the individuals who will benefit from this technology. https://www.cdc.gov/ncbddd/disabilityandhealth/pdf/disabilityposter_photos.pdf